# Antifouling Properties of Silver-Zinc Oxide Polyamide Thin Film Composite Membrane and Rejection of 2-Chlorophenol and 2,4-Dichlorophenol

**DOI:** 10.3390/membranes9080096

**Published:** 2019-08-05

**Authors:** Kate Kotlhao, Isiaka A. Lawal, Richard M. Moutloali, Michael J. Klink

**Affiliations:** 1Faculty of Applied and Computer Sciences, Department of Chemistry, Vaal University of Technology, Private Bag X021, Vanderbijlpark 1900, South Africa; 2Faculty of Science, Department of Applied Chemistry, University of Johannesburg, P.O. Box 17011, Doornfontein, Johannesburg 2028, South Africa; 3Faculty of Applied and Computer Sciences, Department of Biotechnology. Vaal University of Technology, Private Bag X021, Vanderbijlpark 1900, South Africa

**Keywords:** PES, interfacial polymerization, Ag-ZnO/PA-TFC membrane, hydrophilicity

## Abstract

The silver-zinc oxide (Ag-ZnO) polyamide thin film composite (PA-TFC) membrane was prepared by interfacial polymerization. The Ag-ZnO/PA-TFC membrane was characterized by attenuated total reflectance fourier-transform infrared spectroscopy (ATR-FTIR) for polyamide functional groups and contact angle for surface hydrophilicity. The Ag-ZnO/PA-TFC membrane was further characterized by Scanning Electron Microscopy (SEM) and Atomic Force Microscopy (AFM) for morphology and surface roughness, respectively. The performance of the fabricated membrane was investigated using pure water flux, permeability, rejection, flux recovery, and fouling resistance using low molecular weight organic pollutants, 2-chlorophenol (2-CP) and 2,4-dichlorophenol (2,4-DCP). The results were compared to the neat (PA-TFC) membrane. It was observed that incorporation of Ag-ZnO nanocomposites into the PA-TFC membrane improved hydrophilicity, permeation, rejection, and fouling resistance properties of the membrane. The contact angle decreased from 62.8° to 54° for PA-TFC and the Ag-ZnO/PA-TFC membrane, respectively. The presence of Ag-ZnO enhanced permeability of the membrane from 0.9 (Lm^−2^h^−1^bar^−1^) to 1.9 (Lm^−2^h^−1^bar^−1^). Modification of the membrane with Ag-ZnO further showed an enhanced rejection of 2-CP and 2,4-DCP from 43% to 80% and 58% to 85%, respectively. The 2,4-DCP molecules were rejected more than 2-CP due to enhanced repulsive forces from the extra Cl ion. A high flux recovery of about 95% was achieved for the modified membrane compared to 64% for the neat membrane. The improved flux recovery was an indication of enhanced antifouling propensity.

## 1. Introduction

Chlorophenols are listed as priority pollutants (e.g., 2-CP and 2,4-DCP) because of their detrimental effects to water quality, aquatic life, animals, and humans. They disrupt the endocrine system and often lead to endocrine related illnesses [1]. They are usually introduced into water bodies from various sources such as waste from paper and pulp industries, pesticides from agricultural run-off, municipal waste water, and mining industry waste [2]. They occur at low concentrations and often survive even the state-of-the art water treatment processes and systems [3]. As a result, there is a need to develop effective water treatment methods for their reduction or elimination from water.

Membrane technology is preferred due to its unique separation principle [4]. However, membranes suffer an inherent problem of fouling. The resultant drawbacks due to fouling are increased cost, breakdown of equipment, frequent cleaning, membrane degradation, and shorter life span [4,5].

Incorporation of nanomaterials is gaining attention as additives in membranes for water treatment [6]. Nanoparticles possess antibacterial, photocatalytic [7], and hydrophilic properties, which alleviate membrane fouling. Such nanoparticles include zeolite, carbon nanotubes, and photocatalysts (e.g., ZnO, TiO_2_). These materials often provide the membrane with improved mechanical, electrical, optical, chemical, and thermal stability [8,9]. Thin film composite (TFC) membranes incorporated with zeolite showed an enhanced membrane permeability (high flux) and high solute rejection [10,11]. Zeolite nanoparticles are not suitable for rejection of low molecular weight organic compounds with neutral charge [3,12]. Carbon nanotubes (CNTs) have also been incorporated into polyamide thin film composite (PA-TFC) membranes. They have a high specific surface area, high mechanical strength, excellent water transport properties, and chemical inertness [13,14,15]. Lee et al. (2014) prepared a PA-TFC membrane embedded with dopamine coated multi-walled carbon nanotubes (MWCNTs) by interfacial polymerization. They achieved an enhanced permeation [16]. The use of carbon nanotubes is limited in rejection of low molecular weight organic compounds due to the adsorption and desorption mechanisms that lead to greater losses of the analyte [17]. 

Phase inversion and interfacial polymerization are common methods used to incorporate nanoparticles (NPs) into membranes to improve the antifouling properties [18]. The disadvantage of the phase inversion method is the easy desorption of NPs from the membranes. Interfacial polymerization is one of the suitable methods for fabrication of thin film membranes (e.g., PA-TFC) [19]. The structure of the thin film membranes consists of three layers namely; the ultra-thin surface layer, microporous interlayer, and polyester porous layer (support). 

The structure and the morphology of the ultra-thin layer in the PA-TFC membrane has a vital role in rejection of pollutants. Its morphology is a result of the effectiveness of the reaction of the monomers [20]. Solute rejection is also influenced by surface charge of the membrane. It is well understood that charged solutes are rejected by repulsive forces. Li et al. (2014) investigated the effect of different amine monomers, diethylenetriamine triethylenetetramine tetraethylenepentamine (TEPA) and piperazidine (PIP) with an organic soluble monomer, trimesoyl chloride on fabrication of nanofiltration (NF) membranes for rejection of orange GII solution. They observed that PIP/TMC membrane readily rejected the orange GII due to the presence of the negative charge. They concluded that the rejection mechanism was due to repulsive interactions (Donnan effect) [21]. The support material in the PA-TFC membranes also plays an important role in thin film membranes. Polysulfone, polyethersulfone (PES) and polyvinylidene fluoride (PVDF) are common microporous supports. PES is the most advantageous because it is stable at high temperature with high flux properties [22,23].

Zinc oxide (ZnO) and silver (Ag) NPs have been used independently with membranes in water treatment because of their super-hydrophilic (ZnO), antibacterial (Ag) and photocatalytic properties [24,25,26]. Silver NPs have been previously used for its antibacterial properties [27,28]. They have also shown hydrophilic and antifouling properties [29,30]. 

In the current study silver modified ZnO NCs were incorporated into the thin layer of the PA-TFC membrane by interfacial polymerization method. The PES was used as a support material to enhance permeability because of its porous structure. The Ag-ZnO NCs were incorporated to enhance hydrophilicity and to improve the antifouling properties of the PA-TFC membrane. To the best of our knowledge, there is no literature in which the current membrane modification procedure for rejection of 2-CP and 2,4-DCP pollutants was followed. Zinc Oxide NPs was first modified with Ag and incorporated into the PA-TFC membrane. We report synthesis of Ag-ZnO nanocomposites using precipitation and in situ chemical reduction to attach silver to ZnO. The nanocomposites were characterized by FTIR, XRD, SEM, and EDX. These were incorporated into PA-TFC membrane using interfacial polymerization method. The modified membrane was tested for rejection against 2-CP and 2,4-DCP pollutants. Other performance tests included flux, flux recoveries, percentage, and detailed fouling resistance of the membrane; total fouling (reversible and irreversible fouling). 

## 2. Experimental Section

### 2.1. Materials and Methods

Analytical reagent grade chemicals were used without any further purification. Commercial polyether sulfone, 5 kDa was supplied by Microdyn Nadir (Austin, TX, USA). Sodium dodecyl sulphate (98.5%), piperazine (99%), trimesoylchloride (98%), anhydrous hexane (95%), poly (vinyl pyrrolidone) (PVP MW = 10,000 g/mol), zinc nitrate hexahydarte ((Zn(NO_3_)_2_·6H_2_O, 97%), sodium hydroxide (NaOH, 97%), silver nitrate (99%), and sodium borohydride (NaBH_4_, 99%) were obtained from Sigma–Aldrich (Kempton Park, South Africa).

### 2.2. Synthesis of Silver-Zinc Oxide Nanocomposites

Zinc nitrate hexahydrate, Zn (NO_3_)_2_·6H_2_O (7.86 g) and sodium hydroxide (0.79 g) were each dissolved in 50 mL of 0.5 M ethanol. Sodium hydroxide was added dropwise into the solution of zinc nitrate under constant stirring at 4000 rpm for 30 min. Amounts of silver nitrate equivalent to 1 wt%, 3% and 5 wt% ZnO were added and 10 min later sodium borohydride at a ratio of 1:1 to silver nitrate was added to reduce silver ions to silver. Immediately 0.9 g of PVP was added to prevent agglomeration. The reaction mixture was removed from the stirrer after 30 min sealed with aluminum foil and allowed to age for 12 h. The precipitate was centrifuged and washed with a mixture of de-ionized water and ethanol. It was then dried at 80 °C and calcined at 500 °C for 2 h. 

### 2.3. Preparation of Membranes

The procedure followed in the fabrication of PA-TFC membranes using interfacial polymerization was adapted from Mbuli et al. (2017) [19]. Aqueous solutions of piperazine and nanocomposites of Ag-ZnO (0 to 2 wt%), and the organic phase of TMC and hexane were prepared according the quantities shown in Table 1. The aqueous mixture was stirred for 1 h at pH = 8 (adjusted using ammonium chloride) and the organic phase was also stirred for the same duration. To prepare the PA-TFC membrane, commercial PES membrane was firmly adhered to the glass plate using a double sided tape. The aqueous solution was placed covering the surface of the PES support membrane and allowed to soak for 5 min. The excess solution was removed and immediately the organic phase solution was placed over the aqueous phase on the PES for 60 s. After draining excess solution, the membrane was cured in an oven at 65 °C for 15 min to complete polymerization and to form the polyamide layer on the membrane. The fabricated membranes (neat and modified) were washed three times with deionized water then stored wet and cold in deionized water [31]. Scheme 1 is the synthesis reaction of PA-TFC from trimesoyl chloride and pepirazine.

### 2.4. Characterization

Attenuated total reflectance fourier transformed infrared (ATR–FTIR) spectroscopy, Spectrum 100 spectrometer, (PerkinElmer, Shelton, CT, USA) was used to study chemical changes at the membrane surface of PA-TFC and Ag-ZnO/PA-TFC membranes. The FT-IR Perkin Elmer, Spectrum 400 by (Perkin Elmer, Connecticut, USA) was used to characterize the Ag-ZnO NCs. XRD patterns for Ag-ZnO nanocomposites was investigated using Shimadzu-XRD 700, X-Ray Diffractometer (Shimadzu, Tokyo, Japan) Cu Ka radiation (l = 1.154056 Å). Analysis was carried out in the 2θ range from 10°–90°. Morphology of the samples were determined using scanning electron microscopy (SEM) and atomic force microscopy which were performed at Mintek laboratory in Randburg, South Africa and BITRI laboratory in Gaborone, Botswana respectively.

### 2.5. Hydrophilicity

Contact angle measures the hydrophobicity and hydrophilicity of the membrane. The lower the contact angle the higher the hydrophilicity of the membrane surface. Contact angle measurements for pure PES, PA-TFC, and Ag-ZnO (0.5–2.0 wt%)/PA-TFC were determined using DSA 10 Mk2 (Krüss, Germany) equipment. A drop of deionized water from a needle tip was placed on the membrane surface and the magnified image recorded by the digital camera. Measurements were taken 30 s after the drop contacted the membrane surface. At least 10 measurements were taken at different locations on the membrane surface, five of which were used to determine the average contact angle. The contact angles of the water drops were calculated using SCA 20 software (Version 2.04 Build 4). 

### 2.6. Membrane Performance

Pure water flux, permeability (membrane hydraulic resistance), rejection, and flux recovery were employed to determine membrane performance. Pure water flux was investigated using the stirred dead-end cell Sterlitech, HP4750 (Sterlitech, Kent, WA, USA) with a total volume capacity of 300 mL and an active membrane surface area of 14.6 cm^2^. The cell was connected to nitrogen gas cylinder for controlled pressure. A Teflon-coated magnetic stir bar provided a continuous agitation to reduce concentration polarization or “cake” formation, which is a typical problem in dead end cells. The neat PA-TFC and Ag-ZnO/PA-TFC membranes were first compacted for 30 min at 1100 kPa to achieve stabilization. Pure water flux was calculated using Equation (1).
(1)$$J_{wo}=\frac{V}{A\cdot t}$$
where *J_wo_* is the pure water flux (L^.^m^−2.^h^−1^), *V* is the volume, *A* is the effective membrane area (m^2^), and t is the permeation time (h). The average flux was obtained from three replicates.

Permeation flux was calculated from the slope of the fitted linear regression graph of flux against pressure (200 to 800 kPa) [32]. Rejection of 2-CP and 2,4-DCP using neat PA-TFC and Ag-ZnO/PA-TFC was investigated and calculated using Equation (2) [18,33].
(2)$$R\left(\%\right)=\frac{C_{f}-C_{p}}{C_{f}}\times 100\%$$
*C_p_* is the concentration (mg/L) of chlorophenols (2-CP and 2,4-DCP) in the permeate solution and C_f_ is concentration (mg/L) of chlorophenols of the feed solution. Rejection tests were carried out using feed solutions of 2-CP and 2,4-DCP at 5.0 ppm in a dead end cell. The test cell was continuously agitated with a magnetic stirrer to prevent polarization [18]. The residual concentration of the permeate and feed solutions were measured using an ultraviolet spectrophotometer UV-2450 (Shimadzu South Africa (Pty) Ltd., Roodepoort, South Africa) at absorption wavelengths of 280 and 306 cm^−1^ for 2-CP and 2,4-DCP, respectively. The standard solutions (1–5 ppm) were prepared using 1:1 *v*/*v* of chlorophenols and 5.0 M of sodium carbonate solution, respectively [34]. 

#### Antifouling Properties

The membranes were first compacted for 30 min at 1100 kPa to obtain a steady flux. Then, they were subjected to pure water permeation for 60 min to obtain initial water flux (*J_w_*_0_) followed by filtration of 2-CP and 2,4-DCP for another 60 min to obtain water flux (*J_wt_*). Used membranes were cleaned with deionized water for 10 min to remove the foulant molecules loosely deposited on the surface of the membranes. The membranes were re-assessed to obtain flux of the cleaned membranes (*J_wc_*) [18,19]. Water recovery ratio (FRR) were determined using Equation (3);
(3)$$FRR\left(\%\right)=\;\frac{J_{wc}}{J_{w0}}$$
where *J_w_*_0_, *J_wt_*, and *J_wc_* are the water fluxes of pure water, 2-CP/2,4-DCP solutions, and cleaned membranes, respectively. 

Fouling parameters like total fouling ratio (*R_t_*), reversible fouling ratio (*R_r_*) due to loose attachment of foulants on the membrane surface, and irreversible fouling ratio (*R_ir_*) due to adsorption of foulants on membrane pore walls describe the fouling resistance of the membranes. These are represented by Equations (4)–(7) [18].
(4)$$R_{t}\;\left(\%\right)=1-\left[\frac{J_{wt}}{J_{w0}}\right]\;\times \;100\%$$
(5)$$R_{t}=R_{r}+R_{ir}$$
(6)$$R_{r}\left(\%\right)=\left[\frac{J_{wc}-J_{wt}}{J_{w0}}\;\right]\times \;100\%$$
(7)$$R_{ir}\left(\%\right)=\left[\frac{J_{w0}-J_{wc}}{J_{w0}}\right]\times \;100\%$$
A lower *R_t_* value means a better antifouling property, while a higher *FRR* value indicates a higher cleaning efficiency.

### 2.7. Silver Release

Bench-scale batch tests were performed for silver release from the modified Ag-ZnO/PA-TFC membrane. Circular membrane coupons with an area of about 3.0 cm^2^ were placed in 10 mL of 0.9% NaCl (saline water) in a beaker. The purpose of NaCl was to react with silver to give AgCl for detection of Ag^+^ ions. The membranes were continuously agitated using a benchtop orbital shaker for 24 h. The pH was adjusted to pH = 5.0 using HNO_3_ and pH = 8.0 and pH = 10.0 using NaOH. The saline water was collected as a sample every 24 h and a new saline solution was placed in the beakers. The procedure was repeated for six days. The longest contact time according to NSF/ANSI 61 protocol is three days. The leachates were analyzed for Ag using ICP-OES.

## 3. Results and Discussion

### 3.1. Characterization-Nanoparticles

#### 3.1.1. FTIR

The FTIR spectra of Ag, ZnO, and Ag-ZnO are shown in Figure 1. Figure 1(i) is the FTIR spectrum of AgNPs. Figure 1(ii) is the FTIR spectrum of ZnO NPs. The peak at 3022 cm^−1^ was assigned to the CH- stretch. The more intense peak appearing at 1751 cm^−1^ was assigned to the C=O from PVP capping agent. Other peaks from PVP were the C-N stretching vibrations at 1213 cm^−1^ and the doublet peaks at 1458 cm^−1^ (weak) and 1362 cm^−1^ (strong) due to the attached CH_2_ in the pyrrole ring and the ring C-C appearing at 839 cm^−1^ [35,36]. The significant band at 444 cm^−1^ was assigned as the characteristic peak of the Ag stretching mode [37]. From the results, it was evident that the different peaks assigned due to PVP on Ag NPs spectrum appeared weak on ZnO NPs. The results indicate that Ag NPs bind more to PVP than ZnO because upon addition of silver to ZnO, peak intensities characteristic of PVP increased as observed in Figure 1(iii–v) [36]. 

#### 3.1.2. XRD

The room temperature XRD patterns for ZnO and Ag-ZnO are shown in Figure 2. The XRD patterns for neat ZnO exhibited peaks that are characteristic of the hexagonal wurzite structure. All the peaks were in agreement with JCPDS card (No. 36-1451). The diffraction peaks appeared at 2θ = 31.81°, 34.50°, 36.19°, 47.78°, 56.54°, 62.97°, 66.36°, and 68.06° corresponding to (100), (002), (101), (102), (110), (103), (200), and (112) crystalline planes, respectively. Additional low intensity diffraction peaks at 38.2°, 44.4°, 64.7°, and 77.6° corresponding to (111), (200), (220), and (311), respectively, were observed after addition of Ag to ZnO [7]. The hexagonal structure was retained after addition of Ag. A possible explanation is that Ag atoms systematically substituted Zn ions without changing its crystal structure. There was no peak of impurity observed in the XRD pattern, which was an indication of high purity of the ZnO nanoparticles and Ag-ZnO nanocomposites [38,39].

#### 3.1.3. SEM and EDX analysis for Ag, ZnO NPs, and Ag-ZnO NCs

The morphology and size of the nanoparticles and nanocomposites were investigated using SEM. Figure 3 shows SEM images of Ag (a), ZnO (b) nanoparticles, and Ag-ZnO (c) nanocomposites. The images of Ag (Figure 3a) nanoparticles are spherically shaped and mono dispersed with the mean particle size of 68 ± 4 nm. The ZnO nanoparticles predominately appeared as rod-like shaped structures, with random orientations and length ranging from 530 ± 21 nm. Addition of silver to ZnO revealed the presence of spherical particles on the surface of the nanorods, which could be silver NPs. The appearance of the rods in Ag/ZnO nanocomposites and the presence of silver nanoparticles on the surface of the nanorods was also observed by Song et al. (2010). This was attributed to the use of organic solvents, which tend to promote formation of rods with ZnO [40]. In this study methanol was used as the solvent. Zinc oxide crystals have a fast growing 0001 plane during the initial stage of the reaction. Due to this, the plane that elongates more than the others results in the formation of rods [41]. Addition of Ag into ZnO resulted in spherical particles that appeared as agglomerates around the rod shaped ZnO NPs. The estimated length and diameter of ZnO rods was 603 ± 50 nm and 83 ± 5 nm, respectively. The mean sizes for the spherical particles around the rods in Ag-ZnO (3%) was 37 ± 7 nm.

Energy-dispersive X-ray spectroscopy (EDX) was used to determine the percentage elemental composition of Ag in Ag-ZnO and the purity of the nanoparticles and nano-composites synthesized as shown in Figure 4. As observed in the Ag EDX spectrum, the silver element was the only one identified with a percentage weight of 100%. This further agrees with the XRD data, which showed uncontaminated peaks. In the ZnO nanoparticles, EDX spectrum also revealed the presence of Zn and O elements only. For the Ag-ZnO nanocomposites, EDX spectrum showed the presence of Ag, Zn, and O elements only. The EDX spectra for Ag, ZnO nanoparticles, and Ag-ZnO nanocomposites showed no interfering elements just as was observed in XRD, which is an indication of high purity for the prepared nanocomposites.

### 3.2. Membrane Characterization

Commercial membranes such as NF and RO are highly susceptible to fouling with low molecular organic pollutants such as chlorophenols. They also have low membrane permeability that negatively affects the water purification process. The purpose of this section of the research was to fabricate PA-TFC membrane with enhanced antifouling properties by incorporating Ag-ZnO NCs.

#### 3.2.1. FTIR

Figure 5 is the ATR-FTIR spectra of PES, PA-TFC, and Ag-ZnO/PA-TFC (0.5–2.0 wt%) membranes. The PES membrane consists of an aromatic ring and two sulfonyl groups. The typical asymmetric and symmetric S=O bands are expected at 1340–1310 cm^−1^ and 1365–1165 cm^−1^ [42]. The PES spectra confirmed the presence of the asymmetric and symmetric S=O bands at 1316 and 1148 cm^−1^. The C=O (amide I) peak of poly (pepirazine amide) was observed at 1650 cm^−1^. The strong bands appearing at 2924 cm^−1^ and 2846 cm^−1^ region on the PA-TFC and Ag-ZnO/PA-TFC (0.5–2.0 wt%) were due to both the asymmetric and symmetric CH_2_ from pepirazine aliphatic ring. The broad band at 3336 cm^−1^ was assigned as the -OH stretch [43]. The intensity of the –OH group on the membrane surface increased with increasing concentration of the Ag-ZnO nanocomposites. This is because addition of Ag-ZnO NCs increase hydrophilicity [33,44]. The Ag-ZnO NCs within the surface layer promote the attachment of hydroxyl groups. When the percentage weight of the NCs is increased in the thin film layer of PA-TFC membrane, there is a strong interaction with the water molecule through Van der Waals forces and hydrogen bonding leading to the formation of a hydration layer [45]. 

#### 3.2.2. SEM and AFM Analysis

SEM analysis was carried out to investigate surface distribution of Ag-ZnO nanocomposites (NCs) within the PA-TFC membrane. A well modified membrane should show evenly distributed NCs across the surface of the PA-TFC membranes. The cross section of SEM images are shown in Figure 6a,b for PA-TFC and Ag-ZnO/PA-TFC membranes, respectively. The images show dense thin layer on the surface followed by an interlayer and the porous support layer. The thin layer confirms the formation of the PA layer. The results are in agreement with the ATR-FTIR amide I peak observed at 1650 cm^−1^. Figure 6c,d are the SEM images of the surface area of PA-TFC and Ag-ZnO/PA-TFC membranes. The results indicate that the particles of Ag-ZnO are uniformly distributed within the nanocomposite polyamide matrix of the membrane. Both images of SEM and AFM image for surface morphology of PA-TFC and Ag-ZnO/PA-TFC membranes further show the characteristic “ridge-and-valley” structure of polyamide distributed throughout the plane [46]. The AFM image of PA-TFC membrane showed a typical “nodular” feature with a few leaf-like structures compared to Ag-ZnO/PA-TFC membrane. This is because a rougher PA layer is formed on a relatively dense substrate due to an interfacial instability from the curing heat during interfacial polymerization between PIP and TMC [47]. However, surface roughness increased from 12.3 nm to 20.8 nm for PA-TFC and Ag-ZnO/PA-TFC membrane, respectively. The high roughness could increase water flux by providing a larger surface area for water transportation across the membrane. However it could also increase membrane fouling because the foulants easily adsorb to a rough surface. Al Mayyahi et al. (2018) observed similar results from a TiO_2_/PA-TFN membrane fabricated using interfacial polymerization. Where addition of TiO_2_ to the PA-TFN membrane resulted in increased roughness [48].

#### 3.2.3. Surface Hydrophilicity of the Membranes Using Contact Angle

Contact angle measurements were used to evaluate the hydrophilicity of the membrane. Hydrophilic membranes possess a contact angle of 0° < θ < 90°, while hydrophobic membranes have a contact angle of 90° < θ < 180° [49]. Hydrophilicity interaction is described as the hydrogen bonding that occurs between the polar water molecules and the membrane surface. Hydrophilicity of the membrane enhances water flux and antifouling properties of the membrane [50]. Hydrophobic membranes encourage membrane fouling because foulants get adsorbed onto the surface of the membrane forming a cake layer that reduces water flux [51]. The contact angles for modified PA-TFC membranes showed a decrease from 62.8° for neat PA-TFC to 54° for Ag-ZnO/PA-TFC (Figure 7). In our previous work ZnO was incorporated into PES membrane using phase inversion method. The contact angles decreased from 87° to 53° with increasing amount of ZnO nanoparticles on the membrane from 0–2.0 wt% (Figure 7). A corresponding increase in flux was observed [52]. Shen et al. (2012) observed similar results where ZnO was blended into PES membrane and contact angles reduced from 80.0° to 54.7°. Figure 8 shows that beyond 1.5 wt% of Ag-ZnO nanocomposites, contact angle increased. This is because with high concentration of nanomaterials in the aqueous solution, particles tend to agglomerate and affect membrane hydrophilicity [53]. The sudden increase in contact angles beyond 1.5 wt% may also be explained in terms of viscosity effect. That is, increase in the amount of nanoparticles in the aqueous solution increases the viscosity. The viscosity effect dominates and hinders the interfacial polymerization reaction between the aqueous and the organic phase [53].

#### 3.2.4. Water Flux and Permeation Test

To obtain water permeability of the membrane, pure water flux (*J_wi_*) is first obtained and calculated using Equation (1). Hydraulic permeability of the membranes calculated from the slopes of the fitted linear regression on the graphs of water flux against transmembrane pressure [54]. Table 2 are the experimental results of pure water flux at different transmembrane pressure for PES, neat membrane (PA-TFC), Ag-ZnO/PA-TFC nanocomposites. Polyethersulfone (PES) membrane showed the highest pure water permeability compared to both neat PA-TFC and Ag-ZnO/PA-TFC membranes as expected due to its high porosity. However, the presence of Ag-ZnO in the PA-TFC membrane enhanced permeability of the membranes from 0.9 (L^.^m^−2.^h^−1.^bar^−1^) for the neat PA-TFC membrane to 1.9 (L^.^m^−2.^h^−1.^bar^−1^). As in agreement with contact angle results the presence of Ag-ZnO induced hydrophilicity to the PA-TFC membrane surface leading to enhanced permeability and increased flux. 

#### 3.2.5. Rejection Test

Modification of PA-TFC with Ag-ZnO enhanced rejection from 43% to 80% and 58% to 85% for 2-CP and 2,4-DCP, respectively, as shown in Figure 8. The rejection for both 2-CP and 2,4-DCP was attributed to the effects of repulsion between the molecules and the membrane surface [55]. The results showed increased rejection from 80.3% to 85% for 2-CP and 2,4-DCP, respectively. This was attributed to enhanced negative charge due to an extra chlorine that enhanced repulsive forces between the membrane and the 2,4-DCP molecules [56]. At pH above the pKa for 2-CP (8.56) and 2,4-DCP (7.89), the chlorophenols dissociates into a phenolate anions and exist mainly as negatively charged species hence the repulsive forces for rejection [56,57].

#### 3.2.6. Evaluation of Antifouling Properties

Organic fouling is the adsorption or deposition of dissolved organic matter on the membrane surface [58]. This phenomenon ultimately leads to undesirable high energy consumption due to increased pressure requirements [59]. In the present study we assessed the as prepared membrane, Ag-ZnO/PA-TFC for antifouling properties compared to the neat PA-TFC. Figure 9a presents fouling results obtained by alternating flux cycles of pure water with 2-CP or 2,4-DCP (Figure 9b against time). The results indicate that the Ag-ZnO/PA-TFC membrane reached a pure water flux (*J_w_*_0_) of 9.8 and 9.4 L^.^m^−2.^h^−1^, which was higher than the neat PA-TFC membrane at 8.4 and 6.6 L^.^m^−2.^h^−1^ for 2-CP and 2,4-DCP, respectively. The flux reduced in both the neat and modified membranes after replacing pure water with 2-CP and 2,4-DCP. Flux reduction in the PA-TFC membrane was more than in the Ag-ZnO/PA-TFC membrane due to its high susceptibility to fouling. The ability of the modified membrane to exhibit low reduction in flux is an indication of an improved antifouling performance. 

After cleaning the membranes flux recoveries improved from 61% (2-CP) and 64% (2,4-DCP) for neat PA-TFC membranes to 93% (2-CP) and 95% (2,4-DCP) for modified Ag-ZnO/PA-TFC membranes, respectively, as shown in Figure 10. The increase in flux recovery was due to incorporation of Ag-ZnO in each case, which improved the membrane antifouling properties. This means in the case of Ag-ZnO/PA-TFC membrane foulants were only loosely adsorbed at the membrane surface [60,61]. When surface hydrophilicity is high as in the case of modified membrane, the membrane is capable of forming stable hydration layer of hydrogen bonding with water molecules to keep away the organic foulants from being adsorbed on the membrane surface [62,63].

Total fouling (*R_t_*), reversible fouling (*R_r_*), and irreversible (*R_ir_*) for PA-TFC, Ag-ZnO/PA-TFC membranes using 2-CP and 2,4-DCP were computed. The type of fouling in neat PA-TFC for both 2-CP and 2,4-DCP was irreversible and reversible for the neat modified Ag-ZnO/PA-TFC membranes, respectively [18,56,58].

#### 3.2.7. Silver Release

Batch experiments were used to investigate silver leaching from Ag-ZnO/PA-TFC membrane at pH 5, 8, and 10 using NaCl (Figure 11). Figure 12 is a comparison of the Ag-ZnO/PA-TFC control experiment (without NaCl) to the Ag-ZnO/PA-TFC membrane with NaCl, both at pH = 8. The results showed that after the first 24 h the silver concentration for the experiments were lower than at 48 h. This could mean silver in Ag-ZnO was firmly embedded on the membrane surface. In most research silver release test experiment usually show highest levels of silver after 24 h, which they attribute to the loosely adsorbed silver at the surface of the membrane [64,65]. Our results confirm the hypothesis that using interfacial polymerization to embed nanoparticles to the PA layer alleviate the problem of easy desorption of nanoparticles into the water [66]. The highest release was observed at the 48th h after which there was slow release over time. Then silver release was lowest from the 72nd h compared to pH = 5 and pH = 8. This is because at pH > P_izp_ the surface of the ZnO (9.0) is negatively charged and strongly attracted to the Ag^+^ hence small amounts of silver are released. The slow release demonstrates a long-term durability of Ag^+^/Ag available to maintain antifouling properties of the membrane. It also implies that ZnO acts as a good attachment to silver and good in controlling a sustainable Ag release [61]. Upon comparing the Ag-ZnO/PA-TFC membrane in NaCl solution to Ag-ZnO/PA-TFC membrane without NaCl (control, both at pH = 8), it was observed that without NaCl the release of metallic silver was much lower. This is because the solution contains less metallic silver than the silver ions. In the presence of NaCl the silver ions is attracted to Cl ions.

## 4. Conclusions

The purpose of the study was to fabricate Ag-ZnO/PA-TFC membrane with enhanced antifouling properties. Ag-ZnO NCs was successfully prepared using precipitation method incorporated into the PA-TFC by interfacial polymerization (IP) method. The IP method membranes resulted in the formation of polyamide layer shown by the presence of the C=O, amide I group appearing at 1650 cm^−1^. Incorporation of Ag-ZnO nanoparticles showed enhanced hydrophilicity through contact angle measurements compared to the neat PES and PA-TFC membranes. As a result of the hydrophilic enhancement, the as-prepared membranes exhibited good permeation and rejection against 2-CP and 2,4-DCP. The modified membrane showed reversible fouling properties with enhanced flux recoveries. This was an indication of good antifouling properties of the membrane. The Ag-ZnO/PA-TFC showed a steady release of silver, which is a good property for a long lasting antibiofouling membrane.

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
