# Peer review of "Antifouling Properties of Silver-Zinc Oxide Polyamide Thin Film Composite Membrane and Rejection of 2-Chlorophenol and 2,4-Dichlorophenol"

_membranes, 2019, doi:10.3390/membranes9080096_

Round 1

Reviewer 1 Report

The manuscript certainly contains interesting information that fully corresponds to the topic of the journal.
The work was done thoroughly and was written qualitatively. However, it is necessary in all parts of the article, including the abstract, to enter information, how polyamide and a composite based on it appear. It is impossible to prove the presence of an amide bond without explaining in which reaction and from which reagents it appears.
Why did the authors choose polyamide? How does it affect the filtration process and does it change the antimicrobial activity of the nanocomposite? In addition, in the experimental part, the membrane separation method is not described in detail. Why was this method chosen? What happens as a result with the membrane, if known?
 From small remarks - a typo in the inscription on the Fig. 13

Author Response

Reviewer 1:

Comments and Suggestions for Authors

The manuscript certainly contains interesting information that fully
corresponds to the topic of the journal.
The work was done thoroughly and was written qualitatively. However, it
is necessary in all parts of the article, including the abstract, to
enter information, how polyamide and a composite based on it appear.

We have included the schematic chemical equation for the condensation polymerization reaction between trimesoyl chloride and piperazine.

This information has been included in the abstract, methodology, results and conclusion sections

It is impossible to prove the presence of an amide bond without explaining
in which reaction and from which reagents it appears.

The information has been included ( as in the above comment). Trimesoyl chloride and piperizane

Explanation given
Why did the authors choose polyamide?

Explained- line 90-94

How does it affect the filtration process and does it change the antimicrobial activity of the nanocomposite?

Section on antibacterial activity has been removed based on the comment by another reviewer, so explanation on effect of polyamide on antibacterial activity is not provided

 In addition, in the experimental part, the membrane
separation method is not described in detail. Why was this method
chosen?

We understand Membrane separation method to be the rejection method. More explanation of the rejection method has been added

What happens as a result with the membrane, if known?

The explanation of what happens with membrane was discussed in terms of fouling properties. That is reversible fouling and irreversible fouling.

 However, further SEM analysis on fouled membranes was not performed

 From small remarks - a typo in the inscription on the Fig. 13

Now Figure 13 is now Figure 12.

Typo noticed was: (mg/l with one bracket – we have included another bracket

                            X-axis label – Time ( hrs) changed to Time (hr)

Reviewer 2 Report

The focus of the study, i.e removal of chlorophenol compounds from water, is clearly important and the authors have shown some encouraging results using a simple modification of polyamide membranes. The results and the coressponding calculations are well presented in the paper. I believe the paper can be accepted in its current form. 

Author Response

Reviewer 2:

Comments and Suggestions for Authors

Comments and Suggestions for Authors

The focus of the study, i.e removal of chlorophenol compounds from water, is clearly important and the authors have shown some encouraging results using a simple modification of polyamide membranes. The results and the coressponding calculations are well presented in the paper. I believe the paper can be accepted in its current form. 

Thank you

Reviewer 3 Report

The manuscript entitled "Antifouling properties of Ag-ZnO/PA-TFC membrane and rejection of 2-CP and 2,4 -DCP" describes the improvements of the antifouling and antibacterial properties of polymeric membranes. The findings are interesting, and the extensive characterizations are performed to support the conclusions. The topic quite fits the scope of the journal.The referee suggests a minor revision regarding the following concerns. 1- Introduction, the following references are suggested to discuss the antibacterial and antifoulong properties of silver nanoparticles.Design of Antibacterial Poly(ether sulfone) Membranes via Covalently Attaching Hydrogel Thin Layers Loaded with Ag Nanoparticles;A facile approach toward multi-functional polyurethane/polyethersulfone composite membranes for versatile applications;Mussel-inspired chitosan-polyurethane coatings for improving the antifouling and antibacterial properties of polyethersulfone membranes. 2- Figure 5. The authors should indicate the MIC and disc diffusion for the materials against Staphylococcus aureus (S. aureus, Gram positive). 3- Figure 1. The curves of FTIR are overlapped. 4- Some grammar mistakes. For instance, Page 2 line 78. "Sodium dodecyl Sulphate (98.5%), Pepirazine (99%), trimesoylchloride (98%), anhydrous hexane (95%), Poly (vinyl..." should read "Sodium dodecyl sulphate (98.5%), pepirazine (99%), trimesoylchloride (98%), anhydrous hexane (95%), poly (vinyl..."

Author Response

Reviewer 3:

Comments and Suggestions for Authors

The  manuscript entitled "Antifouling properties of Ag-ZnO/PA-TFC membrane  and rejection of 2-CP and 2,4 -DCP" describes the improvements of the  antifouling and antibacterial properties of polymeric membranes. The  findings are interesting, and the extensive characterizations are  performed to support the conclusions. The topic quite fits the scope of  the journal.The referee suggests a minor revision regarding the  following concerns.

1-            Introduction,  the following references are suggested to discuss the antibacterial and  antifoulong properties of silver nanoparticles.Design of Antibacterial  Poly(ether sulfone) Membranes via Covalently Attaching Hydrogel Thin  Layers Loaded with Ag Nanoparticles;A facile approach toward  multi-functional polyurethane/polyethersulfone composite membranes for  versatile applications;Mussel-inspired chitosan-polyurethane coatings  for improving the antifouling and antibacterial properties of  polyethersulfone membranes.

Response  – The section on antibacterial activity was removed as suggested by  another reviewer. Hence only one relevant reference was included. Ref  [30]

2-             Figure  5. The authors should indicate the MIC and disc diffusion for the  materials against Staphylococcus aureus (S. aureus, Gram positive).

Response  –  The section on antibacterial activity  was removed as suggested by  another reviewer hence Figure 5 was removed from the article.

3-     Figure 1. The curves of FTIR are overlapped.

Response – FITR has been replotted

4-  Some grammar mistakes. For instance, Page 2 line 78. "Sodium dodecyl  Sulphate (98.5%), Pepirazine (99%), trimesoylchloride (98%), anhydrous  hexane (95%), Poly (vinyl..." should read "Sodium dodecyl sulphate  (98.5%), pepirazine (99%), trimesoylchloride (98%), anhydrous hexane  (95%), poly (vinyl..."

Response – corrected has suggested.

Reviewer 4 Report

This work studies membrane modification by using Ag-ZnO nanoparticles. It is concluded that the incorporation of Ag-ZnO into PA-TFC membranes improves membrane hydrophilicity, permeation, rejection and fouling resistance. However, there is no much new finding in this work. The data and claims are not well explained and supported. It is not recommended for publication.

1.     Both Ag and ZnO have been used as nanoparticles in membrane fabrication. What is the new ideal of adding them in together? Is there any new improvement from the combination?

2.     What is the structure of the new Ag-ZnO nanoparticle? The characterization from FTIR and XRD only show the existence of both particles, but not the structure of Ag-ZnO.

3.     What is the size of Ag-ZnO?

4.     From Fig 5, does it mean Ag-ZnO has better antimicrobial properties than Ag?

5.     Fig 6, the broad band at 3336 cm-1 is explained by the increase in hydrophilicity due to the addition of Ag-ZnO. How does it increase the –OH stretch?

6.     Fig 8, why does the contact angle increase after 1 wt% of Ag-ZnO addition?

7.     The better rejection of Ag-ZnO/PA-TFC membrane is not clearly explained.

8.     From Fig 10(a), the claimed better antifouling property of Ag-ZnO/PA-TFC membrane is not seen.

Author Response

Reviewer 4:

Comments and Suggestions for Authors

This  work studies membrane modification by using Ag-ZnO nanoparticles. It is  concluded that the incorporation of Ag-ZnO into PA-TFC membranes  improves membrane hydrophilicity, permeation, rejection and fouling  resistance. However, there is no much new finding in this work. The data  and claims are not well explained and supported. It is not recommended  for publication.

1.       Both  Ag and ZnO have been used as nanoparticles in membrane fabrication.  What is the new ideal of adding them in together? Is there any new  improvement from the combination?

Response  – Silver and ZnO have been used independently for improving the  antifouling properties of the membranes with silver usually used for  antibiofouling. There are very few publications if not none on Ag-ZnO modified PA-TFC membranes.  As a result this area has not been adequately researched.  It has been  shown that Ag modified membranes suffer problems of leaching from the  membranes hence unsustainable as an antibiofouling membrane depending on  the method used. Our research initially showed some antibacterial  property of the nanoparticles against E.coli. However, We have since  removed that section to concentrate on the effect of Ag-ZnO on PA-TFC  for enhancing antifouling of the membrane and rejection of 2-CP and  2,4-DCP.  So before engaging on the antibiofouling achievements of the  silver component in the membrane. We discussed the effectiveness of the  Ag-ZnO incorporation on the PA-TFC membrane then its performance for  rejection of target pollutants.

2.       What  is the structure of the new Ag-ZnO nanoparticle? The characterization  from FTIR and XRD only show the existence of both particles, but not the  structure of Ag-ZnO.

Response – the FTIR and XRD discussion  has been rephrased to give more meaningful explanation.

3.       What is the size of Ag-ZnO?

Response –The estimated length of the rods was 603 nm ± 50.4 with diameter of 82.92 ±5.4. The mean sizes for the spherical particles around the rods in Ag-ZnO (3%) was 36.7 ±   6.9 nm.

Response  has been added in the article.

4.       From Fig 5, does it mean Ag-ZnO has better antimicrobial properties than Ag?

Response – section removed

5.       Fig 6, the broad band at 3336 cm-1 is explained by the increase in hydrophilicity due to the addition of Ag-ZnO. How does it increase the –OH stretch?

Response – Explanation provided  

6.       Fig 8, why does the contact angle increase after 1 wt% of Ag-ZnO addition?

Response – beyond  1.5 wt% of Ag-ZnO nanocomposites contact angle increased. This is  because with high concentration of nanomaterials in the aqueous solution  particles tend to agglomerate and affect the hydrophilicity [52]. The  sudden increase in contact angle may also be explained in terms of  viscosity effect. That is, the high amount of the nanoparticles in the  aqueous solution increases the viscosity. Viscosity effect dominates and  hinders interfacial polymerization between the aqueous and the organic  phase and affect hydrophilicity

Explanation added in the artiocle

7.       The better rejection of Ag-ZnO/PA-TFC membrane is not clearly explained.

Response – more explanation included

8.       From Fig 10(a), the claimed better antifouling property of Ag-ZnO/PA-TFC membrane is not seen.

Response – More explanation given

Reviewer 5 Report

1.       Abstract: The abstract should be shortened to only emphasize the gist of the article.

2.       Better to use L m-2h-1bar-1 as a unit of flux.

3.       Line 104, the spelling of “pepirazine” is wrong.

4.       Page 4 (and throughout the text), the size and format of texts in the equations must be unified.

5.       Figure 2 is blurring and unpleasant, please increase resolution.

6.       Figure 9. The standardized curve between absorbance and concentration should be moved to supporting information.

7.       What is the mechanism behind the enhanced anti-fouling properties of Ag-ZnO composite membrane against 2-CP and 2-DCP?

8.       The addition of Ag to ZnO is not directly related to the antifouling or rejection properties (as suggested by the title), hence it seems a little off the topic that the authors include the antibacterial properties of the Ag/ZnO materials (but not the membrane) in the manuscript. The authors can consider remove the antibacterial part, or include new information regarding the antibacterial test for the Ag/ZnO composite membrane.

9.       There is no comparison to other piperazine based nanocomposite membranes.

10.    Figure 7. The FESEM image of the Ag-ZnO does not seem to show successful blending of Ag/ZnO.

11.    There seems to be a contradiction as to where the author blended the Ag-ZnO. Was it bended in the PSF support or in the piperazine aqueous solution?

Line 14-Line 17:

“The Ag-ZnO NCs were characterized using UV-Vis, XRD, SEM and EDX. They were investigated for antimicrobial activity using MIC and disc diffusion against E.coli. The composites (varied amounts from 0.0 wt % -2.0 wt%) were blended into a polyether sulfone membrane (support) to fabricate”

Line 103-105: “Aqueous solutions of pepirazine and nanocomposites of Ag-ZnO (0 to 2 wt%), and…”

The grammar needs to be re-checked as there are some casual mistakes.

Author Response

Reviewer 4:

Comments and Suggestions for Authors

1.      Abstract: The abstract should be shortened to only emphasize the gist of the article.

Response – the abstract has been shortened, aspects on antimicrobial activity removed. It now focused on antifouling performance of the modified membrane and rejection of chlorophenols.

2.      Better to use L m-2h-1bar-1 as a unit of flux.

Response- Amended as per comment

3.      Line 104, the spelling of “pepirazine” is wrong.

Response – pepirazine has been changed to piperazine

4.      Page 4 (and throughout the text), the size and format of texts in the equations must be unified.

Response – the text and the equation has been unified as suggested.

5.      Figure 2 is blurring and unpleasant, please increase resolution.

Response – figure 2 has been adjusted

6.      Figure 9. The standardized curve between absorbance and concentration should be moved to supporting information.

Response - The standardized curve between absorbance and concentration has been removed.

7.      What is the mechanism behind the enhanced anti-fouling properties of Ag-ZnO composite membrane against 2-CP and 2-DCP?

Response – Explanation given

8.      The addition of Ag to ZnO is not directly related to the antifouling or rejection properties (as suggested by the title), hence it seems a little off the topic that the authors include the antibacterial properties of the Ag/ZnO materials (but not the membrane) in the manuscript. The authors can consider remove the antibacterial part, or include new information regarding the antibacterial test for the Ag/ZnO composite membrane.

Response – Section on antibacterial activity has been removed

9.      There is no comparison to other piperazine based nanocomposite membranes.

Response – literature not available

10.   Figure 7. The FESEM image of the Ag-ZnO does not seem to show successful blending of Ag/ZnO.

Response – we explained this as silver being distributed around the ZnO rods

11.   There seems to be a contradiction as to where the author blended the Ag-ZnO. Was it bended in the PSF support or in the piperazine aqueous solution?

Line 14-Line 17:

“The Ag-ZnO NCs were characterized using UV-Vis, XRD, SEM and EDX. They were investigated for antimicrobial activity using MIC and disc diffusion against E.coli. The composites (varied amounts from 0.0 wt % -2.0 wt%) were blended into a polyether sulfone membrane (support) to fabricate”

Line 103-105: “Aqueous solutions of pepirazine and nanocomposites of Ag-ZnO (0 to 2 wt%), and…”

Response – Corrected. Sentence removed in line 14

The grammar needs to be re-checked as there are some casual mistakes.

Response – the grammar has been checked and correction make where necessary.

Round 2

Reviewer 4 Report

The revised work still fails to address the issue of innovation. Both Ag and ZnO have been used as nanoparticles in membrane fabrication. Their effects on membrane hydrophilicity and anti-fouling properties have also been largely studied. What is the main difference between this work and previous studies? What is the advantage of adding the combined Ag-ZnO nanoparticle? Is there any new improvement and observation?

Author Response

The revised work still fails to address the issue of innovation. Both Ag and ZnO have been used as nanoparticles in membrane fabrication. Their effects on membrane hydrophilicity and anti-fouling properties have also been largely studied. 

What is the main difference between this work and previous studies?

Response

It has never been done following the same procedure

 What is the advantage of adding the combined Ag-ZnO nanoparticle?

Response: 

To find out if the two will still show antifouling properties. We will  then study both the anti biofouling  and anti organic fouling properties

 Is there any new improvement and observations?

Response

Addition of Ag-ZnO has shown improved antifouling properties using 2-CP and 2,4 -DCP

Response

There has never been a report on incorporation of Ag-ZnO nanoparticles on PA-TFC for rejection of 2-CP and 2,4 DCP  following the same procedure 

Reviewer 5 Report

improved.

Author Response

Some improvements have been added on the:

Introduction

Spelling mistakes corrected

Gramma has been corrected

large spaces have been  reduced

Font and size on graph labels and scale have been  made uniform